# Blood Lead Concentrations and Mortality in Korean Adults: the Korea National Health and Nutrition Examination Survey with Mortality Follow-Up

**DOI:** 10.3390/ijerph17186898

**Published:** 2020-09-21

**Authors:** Garam Byun, Sera Kim, Soo-Yeon Kim, Dahyun Park, Min-Jeong Shin, Hannah Oh, Jong-Tae Lee

**Affiliations:** 1Department of Public Health Science, Graduate School, Korea University, Seoul 02841, Korea; garam0110@gmail.com (G.B.); inb14@naver.com (S.-Y.K.); 2Interdisciplinary Program in Precision Public Health, Korea University, Seoul 02841, Korea; ssera0905@gmail.com (S.K.); ekgus7171@naver.com (D.P.); mjshin@korea.ac.kr (M.-J.S.); hannahoh@korea.ac.kr (H.O.); 3School of Biosystems and Biomedical Sciences, College of Health Science Korea University, Seoul 02841, Korea; 4School of Health Policy and Management, College of Health Science, Korea University, Seoul 02841, Korea

**Keywords:** blood lead, mortality, cancer, KNHANES

## Abstract

Previous studies have consistently reported an increase in mortality risk, even at low levels of blood lead. The average blood lead concentration in the Korean population has steadily decreased but is still higher than that of developed countries. The purpose of this study was to examine the associations between mortality and blood lead concentrations for adults in Korea. We used the Korea National Health and Nutrition Examination Survey (2008–2013) linked Cause of Death data, which are followed by 2018. A total of 7308 subjects who aged over 30 at the baseline examination were included in the analyses. Cox proportional hazard model was used to estimate the hazard ratios of mortality from non-accidental causes and cancer mortality. The estimated hazard ratios (95% CI) for comparison of the second and third tertile group with the lowest tertile group were 2.01 (1.20, 3.40) and 1.91 (1.13, 3.23) for non-accidental mortality and 3.42 (95% CI: 1.65, 7.08) and 2.27 (95% CI: 1.09, 4.70) for cancer mortality, respectively. The dose–response relationship also showed significant increase in the risk of mortality at blood lead level between 1.5 and 6.0 μg/dL. Our findings suggest that potent policies to lower lead exposure are required for the general Korean population.

## 1. Introduction

Lead, a toxic heavy metal found throughout the environment, is known to have various health effects when exposed to the human body. Environmental exposure to lead mainly results from ingestion or inhalation of contaminated dust, water, and food. The adverse impact of lead ranges from neurological and cognitive disorders to kidney or cardiovascular system damage [1]. Epidemiological studies have shown that lead exposure is even linked with mortality. Several studies based on the U.S. National Health and Nutrition Examination Surveys (NHANES) have reported increased risk in mortality from all causes, circulatory diseases, or cancer at relatively low levels of blood lead [2,3,4,5,6,7,8]. For these reasons, lead is still an important public health concern, even though the lead exposures have decreased sharply over recent decades, thanks to the leaded gasoline regulation in developed countries including the United States.

The Centers for Disease Control and Prevention (CDC) had a “level of concern” of 10 μg/dL for blood lead levels (BLLs) in children until 2012. However, on the ground of the growing number of scientific studies, the CDC concluded that no safe level exists in BLL and established a “reference level” of 5 μg/dL. This value represents the 97.5th percentile in the BLL distribution for children of the NHANES [9]. The reference level for adults has also been designated as 5 μg/dL by the National Institute for Occupational Safety and Health (NIOSH) in 2015 [10,11].

The BLL in South Korea (hereafter, Korea) has also steadily decreased since the early 1990s when the leaded gasoline was banned, reaching around 2 μg/dL recently [12]. Nonetheless, this is still higher than the average BLL of 0.8 μg/dL in the U.S [13]. To the best of our knowledge, however, the association between BLL and mortality has never been examined in the general population of Korea. Some studies of the Korean population have suggested the associations between BLL and hypertension, cardiovascular diseases, or metabolic syndrome [14,15,16,17,18,19], based on a cross-sectional design. Kim et al. [20] were the only ones to investigate the relationship between BLL and mortality in Korea, however, it is difficult to generalize their results because the study subjects only included lead-exposed workers.

Ingestion of lead-contaminated food is one of the major pathways of lead exposure. Foods with a high contribution to lead exposure may vary depending on the local environmental condition or the dietary pattern of the population. In Korea, it was reported that grains, vegetables, and seafood are an important contributor to lead intake [21]. As dietary pattern is a well-known risk factor of mortality, high-lead-containing food intake can act as a potential confounder in the association between BLL and mortality. Yet, dietary factors are not commonly considered as a confounding factor in previous studies on BLL and mortality.

The Korea National Health and Nutrition Examination Survey (KNHANES), an ongoing national cross-sectional survey conducted by the Korean Center for Disease Control (KCDC), is designed to assess the health and nutritional status of the representative population of Korea. Recently, KCDC linked the KNHANES (2007–2015) with data of death certificates registered in Statistics Korea. Using this database, we aimed to examine the associations of BLL with risk of death for Korean adults, considering dietary lead exposure as a potential confounder.

## 2. Materials and Methods

### 2.1. Data

We obtained the Korea National Health and Nutrition Examination Survey linked Cause of Death data (ver 1.1) from KCDC and Statistics Korea. The KNHANES has collected data by health interview, health examination, and nutrition survey using a complex, multi-stage probability sample design. The survey has been described in detail elsewhere [22]. Around 95% of the individuals who participated in the KNHANES 2007–2015 were linked to mortality data from death certificates by 2018. The linkage of the data was made using the resident registration number with the consent of the participants. Among these mortality follow-up data for the KNHANES participants, the present study used the KNHANES 2008–2013, when the blood heavy metals were measured.

The measurement of heavy metals in blood was conducted by selecting part of the subjects in the health examination: 2000 people per annum aged 20 or older in 2008–2009 and 2400 people per annum aged 10 or older in 2010–2013. Blood samples were collected in standard commercial evacuated tubes containing sodium heparin (Vacutainer; Becton, Dickinson & Co, Franklin Lakes, NJ, USA). BLL was measured by graphite-furnace atomic absorption spectrometry with Zeeman background correction (Analyst 600, Perkin Elmer, Finland). The analyses were performed at the Neodin Medical Institute, a laboratory certified by the Korean Ministry of Health and Welfare. The limit of detection (LODs) was 0.12 μg/dL and none of the samples were below the LOD. For internal quality assurance and control, commercial reference materials were used (Lyphochek^®^ Whole Blood Metals Control, Bio-Rad Laboratories, Hercules, CA, USA). The coefficients of variation, calculated monthly from samples, were less than 10% and did not exceed the allowable range. For external quality assurance and control, the Neodin Medical Institute passed both the German External Quality Assessment Scheme operated by Friedrich-Alexander University and the Quality Assurance Program operated by the Korea Occupational Safety and Health Agency.

This study was limited to individuals with a BLL less than 10 μg/dL, who were aged 30 years and over at the baseline examination, and who were not diagnosed with cancer or ischemic heart disease. We excluded individuals whose information was lacking for the following covariates. Sex was categorized as males and females. Age at baseline was categorized as 30–44, 45–64, and ≥65 years. The household income was calculated by dividing total household income by the square root of the numbers of household members and then was categorized into quartiles. Education level was classified into 4 categories: elementary school or lower, middle school, high school, and college or higher. Occupations were classified into non-manual (white-collar), manual (blue-collar), and unemployed or housewives. Smoking status was categorized as never-smoker, former smoker, and current smoker. Drinking frequency was categorized as non-drinker, once in a month or less, and more than once in a month. Body mass index (BMI) was divided into <18.5, 18.6–22.9, 23–24.9, and ≥25 kg/m^2^. Levels of physical activity were classified into two categories: participating in moderate- or vigorous-intensity activities more than once a week and not participating in any such activities. Finally, we considered high-lead-containing food intake as a potential confounding factor. The groups of high-lead-containing food have been selected for grains, vegetables, and seafood, which are known to contribute the most to lead exposure by dietary intake in the Korean population [21]. The total amount of daily intakes of selected food groups calculated from 24hr recall was categorized into quartiles. Those who had extremely low or high total energy intake (<500 kcal or >5000 kcal) were also excluded from the study, resulting in a total of 7308 subjects (57,243 person years) to be analyzed (Figure 1).

We identified deaths from all non-accidental causes (the International Classification of Disease tenth revision: ICD-10, A00-R99) and cancer (ICD-10, C00–97). Death from circulatory diseases (ICD-10, I00-99) could not be analyzed due to a lack of death counts. Those who were not matched to death certificate records or were considered censored on December 2018. For cause-specific analyses, those who died from other causes were considered censored at the time of death.

This study was approved by the Institutional Review Board of Korea University (IRB no. KUIRB-2020-0096-01).

### 2.2. Statistical Analysis

The hazard ratios (HRs) of mortality associated with the tertile of BLL were estimated from Cox proportional hazards model. The tertile limits of BLL were 1.91 and 2.71 μg/dL. Survival time was defined as the difference in months between the health examination date and the death date or the end of follow-up (December 2018), since the date of examination and death was recorded using the year and month of the date. We stratified the baseline hazard by examination year and region of residence (16 administrative districts). Initial models (Model 1) were adjusted only for age and sex. Subsequent models (Model 2) were additionally adjusted for household income, education, occupation, smoking status, drinking frequency, BMI, and physical activity. Final models (Model 3) were further adjusted for intake of high-lead-containing food intake (grains, vegetables, and seafood). We tested the proportional hazard assumption using scaled Schoenfeld residuals.

To assess the dose–response relationship between BLL and the risk of mortality, we used a natural cubic spline with three degrees of freedom. The Cox regression spline models were adjusted for all covariates included in Model 3.

In the data construction stage, SAS 9.4 (SAS Institute, Cary, North Carolina, the United States) was used. For statistical analyses, R software 3.5.3 (The R Development Core Team, Vienna, Austria) was used. All analyses were performed with the survey package in R to deal with the complex sampling design and weights in KNHANES.

## 3. Results

Figure 2 shows the annual trend in the distribution of BLL of the study subjects for the period between 2008 and 2013. During the study period, the average BLL steadily decreased. The geometric mean of the BLLs was 2.26 (± 1.52) μg/dL for the study subjects, and 2.8 percent of the total subjects exceeded the reference value (5 μg/dL) of CDC.

Table 1 provides the characteristics of the study subjects. During the follow-up, 205 deaths from non-accidental causes and 87 deaths from cancer were identified. BLLs were higher for male, older groups, those with low income and education level, smokers, those who drink more than once a month, those who are overweight or obese, and those who participate in physical activities. BLL also increased as the daily intake of high-lead-containing food increased.

Table 2 presents the estimated HRs of mortality from non-accidental causes and cancer by tertile of BLL from three different adjustment models. Overall, we found a positive association between BLL and mortality. In Model 2, where basic confounders were adjusted, HRs for the 2nd and 3rd tertile versus the lowest tertile were 2.01 (95% CI: 1.20, 3.40) and 1.91 (95% CI: 1.13, 3.23) for non-accidental mortality and 3.42 (95% CI: 1.65, 7.08) and 2.27 (95% CI: 1.09, 4.70) for cancer mortality, respectively. Adjustment of high-lead-containing food intake did not significantly change the HR estimates (Model 3).

The dose–response relationship between BLL and mortality was investigated with a natural cubic spline. The results are shown graphically in Figure 3, displayed as HR and 95% CI using a reference value of 0.5 μg/dL. The HR increased without threshold but decreased after around 5 μg/dL of BLL in non-accidental mortality and around 2.5 μg/dL of BLL in cancer mortality. The confidence intervals, however, were considerably wide especially for both ends of the x-axis due to the small sample size, making it difficult to reliably interpret the relationship. Nevertheless, the increase in risk of mortality was noticeable at BLL between around 1.5–6.0 μg/dL.

## 4. Discussion

In this cohort-based study on adults representative of Korea, we found a significant increase in the risk of non-accidental and cancer mortality related to higher BLL. In dose–response relationship analyses, elevated risk of mortality was observed even at the lower-than-average BLL without threshold effect, strengthening the hypothesis that BLL is a risk factor of mortality.

Previous research using NHANES with mortality follow-up data in the U.S repeatedly observed the associations of BLLs with mortality due to all-cause and cardiovascular diseases [5,6,8]. In addition, several studies found that a substantially low BLL, which is similar to the levels observed in our study, was associated with mortality [5,7]. The results of the present study are consistent with these studies, especially in all-cause mortality, although the estimated effect size is relatively large. Lanphear et al. [5], for example, reported a HR of 1.37 (95% CI, 1.17–1.60) and 1.70 (95% CI, 1.30–2.22) for non-accidental and circulatory mortality, respectively, for an increase in BLL from 10th to 90th percentiles in the U.S. adults. Menke et al. [7] assessed the mortality effects of BLLs below 10 ug/dL and showed HR of 1.25 (95%CI, 1.04–1.51) for non-accidental mortality in the highest tertile group (≥3.63 ug/dL). Besides studies with NHANES data, other population-based studies also reported that increased risks of death from non-accidental causes and coronary heart disease in women with higher BLLs [23]. For cancer mortality, however, the results are inconsistent. While Lustberg and Silbergeld [6] and Schober et al. [8] found an increase in risk of cancer mortality at the different range of BLLs, from 5–9 ug/dL to 20–29 ug/dL, Jemal et al. [4] and Menke et al. [7] did not find an apparent association of cancer mortality with BLLs below 10 ug/dL.

Related studies conducted in Korea also support the adverse health effects of BLL, although most of them are limited to a cross-sectional design. A cohort study including lead-exposed workers found evidence that the median BLLs were related to increased mortality from all-cause and total cancer [20]. Cross-sectional studies using KNHANES found associations between BLL and high blood pressure [14,15], cardiovascular diseases [19], and metabolic syndrome [17,18] in the Korean adult population.

Our dose–response relationship analyses showed that the risk of mortality increased without threshold, although the curve flattened out at high concentrations. Using NHANES II (1976–1980), Jemal et al. [4] identified the relationship between BLL and mortality risk only in women, with a threshold of 24 μg/dL. A study of NHANES III (1988–1991) reported that the mortality risk became elevated from around 3 μg/dL of BLL and was statistically significant at over 5 μg/dL [8]. Analysis of NHANES (1988–1994) found increased mortality at BLL above 2.0 μg/dL [7]. More recent findings have shown no clear threshold in the relationship of BLL and mortality or suggested that there is a linear relationship [2,5,24]. With regard to the tendency to lower threshold values in recent studies compared to older studies, Schober et al. [8] speculated that the BLL measured in older studies might have overestimated the actual BLL of the population at that time. We have found no clear biological explanation for the threshold effect or the phenomenon of varying slopes at higher concentrations (as shown in this study) in the dose–response relationship. Since we firstly investigated the dose–response relationship between BLL and risk of death in the general population of Korea, replication of this relationship would be needed to produce more reliable results.

Plausible biological mechanisms between lead exposure and health effects have been suggested from numerous experimental and epidemiologic studies. Several studies suggested that lead exposure is related to high blood pressure even at low levels [25,26,27,28]. Lead exposure causes some biological responses related to blood pressure such as oxidative stress [29], inhibition of endothelial nitric oxide synthase [30,31,32], alteration of calcium homeostasis [33,34] or renal damage [35,36]. Ultimately, these responses induce an increase in blood pressure that results in circulatory diseases, including hypertension and atherosclerosis [37]. There are also plausible mechanisms between lead exposure and cancer-related health effects. Lead exposure can replace zinc in DNA binding proteins that lead to an inhibition of DNA repair and induce oxidative stress that contributes to DNA damage [29,38,39]. Some studies suggested that lead exposure might enhance the genotoxicity when co-exposed with other carcinogens like tobacco smoke or ultraviolet radiation [40,41,42].

The present study considered a high-lead-containing food intake as a potential confounder. Food is a major pathway through which humans are exposed to lead in the environment and, at the same time, directly or indirectly affects health status through a variety of nutrients. Seafood, including fish, generally have the highest lead content, while grains and vegetables have lower lead content, but their high amount of consumption contributes to high lead exposure [22]. If the nutrients contained in this food group have a healthy effect, consumption of these food groups is likely to act as a negative confounder in the link between lead exposure and mortality. In our Cox regression analyses, additional adjustment of high-lead-containing food intake did not significantly change the effect estimates overall. Although this study did not observe an apparent confounding effect of high-lead-containing food intake, measurement errors are likely to exist because daily food intake was measured by 24hr recall, which was conducted once at the survey. The food intake calculated from 24hr-recall method might not reflect the usual dietary intake pattern.

The present study has several limitations. First, it is unclear whether the increased mortality observed in this study is associated with the acute effect or chronic effect of lead. Blood lead is known to be more related to current exposure rather than cumulative exposure, and bone lead is often regarded as an indicator of the cumulative lead exposure [43]. Blood lead and bone lead could be positively correlated, however, the results of studies on the link between these two biomarkers for lead exposure and health are inconsistent. While Glenn et al. [44] and Lee et al. [45] have shown that lead levels in both blood and bone are related to high blood pressure, Cheng et al. [46] observed that only bone lead is significantly associated with hypertension. In addition, we may have exposure misclassification as BLL was measured only once during the survey. It also should be noted that the follow-up period was relatively short, with an average of 7.8 years per person, resulting in a small number of deaths identified. Due to the limited sample size, deaths from circulatory diseases were excluded from the main outcome of interest, and tests for effect modification by individual characteristics have not been made. Previous studies have reported that characteristics including age, sex, smoking status, or dietary calcium could be potential effect modifiers of the relationship between lead and health outcomes [4,5,47]. Finally, there still remains the possibility of residual confounding by occupational exposure to lead or other unmeasured factors. Yet, we attempted to address well-known confounders including several socioeconomic status and health behaviors.

Despite these limitations, our study for the first time provided epidemiological evidence for mortality effects of BLL in adults based on KNHANES, a nationally representative sample of the Korean population. Our findings support the hypothesis that BLL is a risk factor for death even at low concentrations. The KNHANES-linked Cause of Death data will be updated every year and further studies based on updated data will be needed to build up the evidence in Korea. We expect the KNHANES to be able to extend the subjects of blood heavy metal measurement to children, and to monitor cumulative lead exposure through bone lead measurement.

## 5. Conclusions

The present study supports the body of evidence on lead exposure and health effects. We observed that BLLs are associated with an increased mortality in Korean adults. The risk of mortality might be elevated even at relatively low BLL, but the results should be interpreted with caution because of the small number of death cases. While the BLL in most of the study population was under the CDC reference value, continuous efforts will be needed to reduce the BLL in the general population, such as strengthening the standard for lead in food or ambient air. Policies to manage the BLL will contribute to achieving the Sustainable Development Goal target (3.9 By 2030, substantially reduce the number of deaths and illnesses from hazardous chemicals and air, water and soil pollution and contamination) in the long run.

## Figures and Tables

**Figure 1 ijerph-17-06898-f001:**
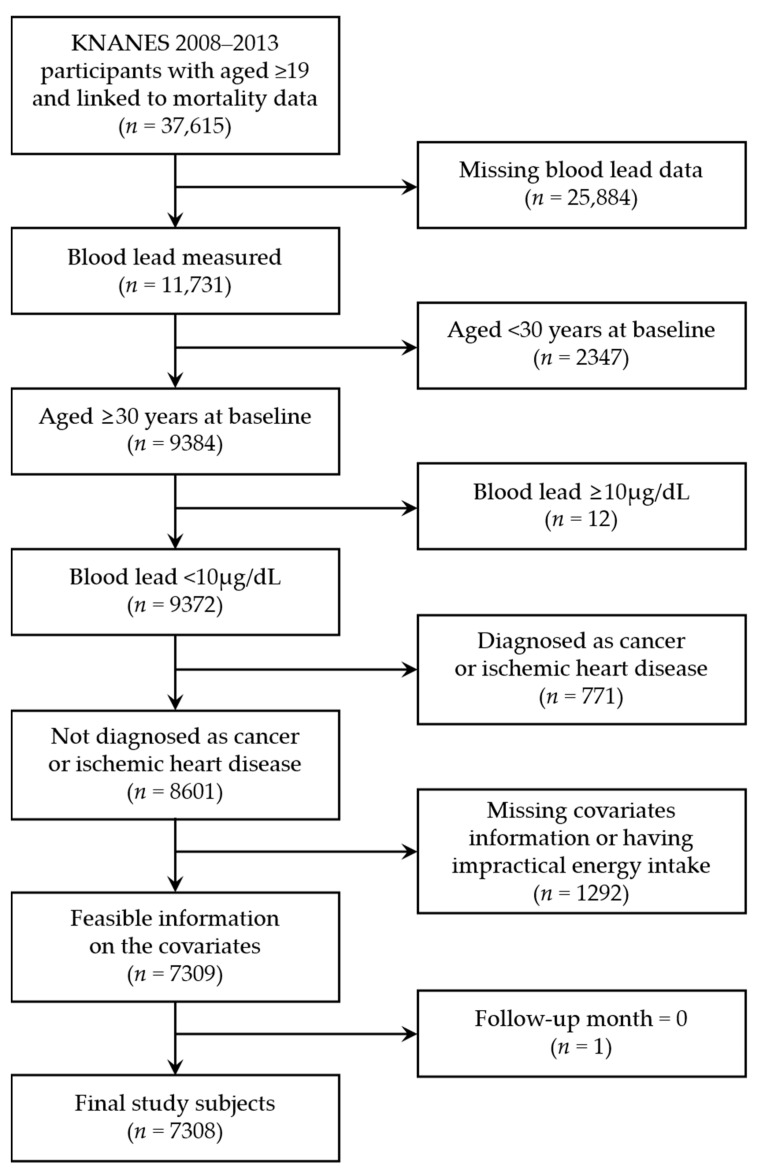
Flow chart of the selection process of study subjects from the Korea National Health and Nutrition Examination Survey.

**Figure 2 ijerph-17-06898-f002:**
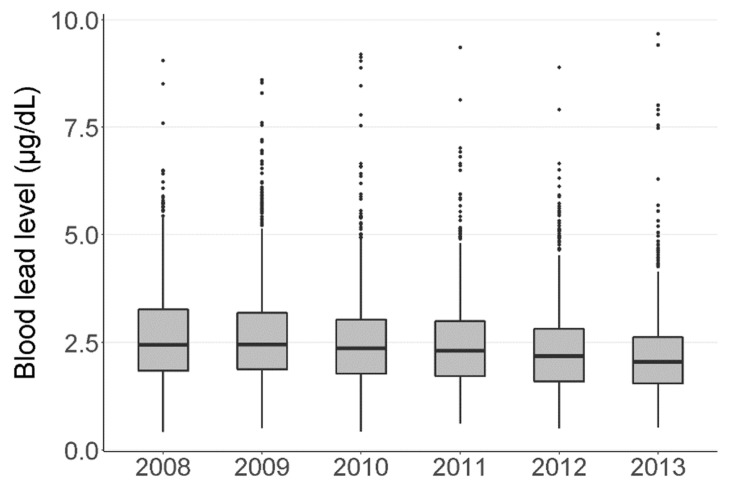
Distributions of blood concentrations in the study subjects, Korea National Health and Nutrition Examination Survey (KNHANES) 2008–2013.

**Figure 3 ijerph-17-06898-f003:**
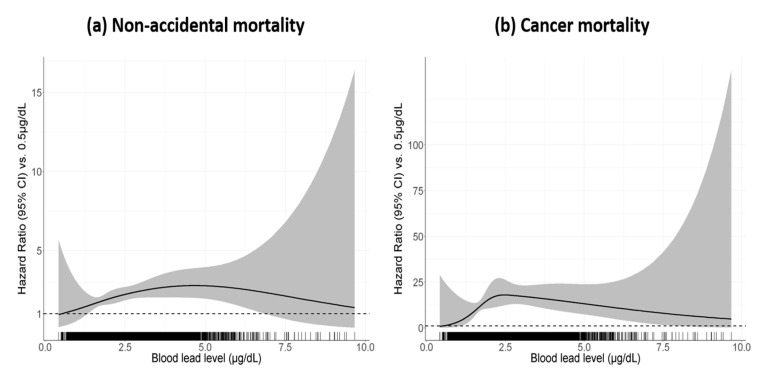
The hazard ratio of (**a**) non-accidental and (**b**) cancer mortality for blood lead levels compared with referent level of 0.5 μg/dL. The solid lines represent the fitted spline relationship and the shaded areas represent the 95% CI. The tick marks on the *x*-axis identify the location of the blood lead concentrations.

**Table 1 ijerph-17-06898-t001:** Baseline characteristics of study subjects by blood lead level, *n* (%).

Characteristics	Total	Blood Lead Level (μg/dL, Tertiles)		
		<1.91	1.91–2.71	>2.71
Non-accidental death				
No	7103 (97.2)	2357 (98.5)	2397 (97.4)	2349 (95.7)
Yes	205 (2.8)	37 (1.6)	63 (2.6)	105 (4.3)
Cancer death				
No	7221 (98.8)	2381 (99.5)	2427 (98.7)	2413 (98.3)
Yes	87 (1.2)	13 (0.5)	33 (1.3)	41 (1.7)
Sex				
Male	3380 (46.3)	537 (22.4)	1146 (46.6)	1697 (69.2)
Female	3928 (53.8)	1857 (77.6)	1314 (53.4)	757 (30.9)
Age (years)				
30–44	2890 (39.6)	1293 (54.0)	952 (38.7)	645 (26.3)
45–64	3408 (46.6)	815 (34.0)	1187 (48.3)	1406 (57.3)
≥65	1010 (13.8)	286 (12.0)	321 (13.1)	403 (16.4)
Household income				
Low	1189 (16.3)	309 (12.9)	397 (16.1)	483 (19.7)
Lower-middle	1931 (26.4)	616 (25.7)	636 (25.9)	679 (27.7)
Upper-middle	2079 (28.5)	748 (31.2)	677 (27.5)	654 (26.7)
High	2109 (28.9)	721 (30.1)	750 (30.5)	638 (26.0)
Education				
≤Elementary	1619 (22.2)	426 (17.8)	520 (21.1)	673 (27.4)
Middle school	926 (12.7)	200 (8.4)	312 (12.7)	414 (16.9)
High school	2522 (34.5)	833 (34.8)	867 (35.2)	822 (33.5)
≥College	2241 (30.7)	935 (39.1)	761 (30.9)	545 (22.2)
Occupation				
Non-manual	1542 (21.1)	558 (23.3)	552 (22.4)	432 (17.6)
Manual	3240 (44.3)	811 (33.9)	1081 (43.9)	1348 (54.9)
Unemployed	2526 (34.6)	1025 (42.8)	827 (33.6)	674 (27.5)
Smoking status				
Never	4328 (59.2)	1899 (79.3)	1468 (59.7)	961 (39.2)
Former	1350 (18.5)	242 (10.1)	457 (18.6)	651 (26.5)
Current	1630 (22.3)	253 (10.6)	535 (21.8)	842 (34.3)
Drinking frequency				
None	1872 (25.6)	770 (32.2)	634 (25.8)	468 (19.1)
≤Once in a month	2136 (29.2)	929 (38.8)	720 (29.3)	487 (19.9)
>Once in a month	3300 (45.2)	695 (29.0)	1106 (45.0)	1499 (61.1)
Body mass index (kg/m^2^)				
≤18.5	216 (3.0)	101 (4.2)	63 (2.6)	52 (2.1)
18.6–22.9	2835 (38.8)	1067 (44.6)	912 (37.1)	856 (34.9)
23–24.9	1840 (25.2)	526 (22.0)	633 (25.7)	681 (27.8)
≥25	2417 (33.1)	700 (29.2)	852 (34.6)	865 (35.3)
Physical activity				
<Once in a week	3487 (47.7)	1252 (52.3)	1128 (45.9)	1107 (45.1)
≥Once in a week	3821 (52.3)	1142 (47.7)	1332 (54.2)	1347 (54.9)
High-lead-containing food intake ^1^				
1st quartile	1825 (25.0)	729 (30.5)	616 (25.0)	480 (19.6)
2nd quartile	1827 (25.0)	624 (26.1)	621 (25.2)	582 (23.7)
3rd quartile	1827 (25.0)	562 (23.5)	623 (25.3)	642 (26.2)
4th quartile	1829 (25.0)	479 (20.0)	600 (24.4)	750 (30.6)
Total	7308 (100.0)	2394 (32.8)	2460 (33.7)	2454 (33.6)

^1^ Grains, vegetables, and seafood.

**Table 2 ijerph-17-06898-t002:** Hazard ratio (95% CI) of non-accidental and cancer mortality associated with tertile of blood lead.

Cause of Death/Tertile of Blood Lead	Model 1	Model 2	Model 3
	Hazard Ratio (HR) (95% CI)	HR (95% CI)	HR (95% CI)
Non-accidental			
1st tertile	reference	reference	reference
2nd tertile	2.09 (1.25, 3.49)	2.01 (1.20, 3.37)	2.02 (1.20, 3.40)
3rd tertile	2.09 (1.27, 3.44)	1.93 (1.14, 3.25)	1.91 (1.13, 3.23)
Cancer			
1st tertile	reference	reference	reference
2nd tertile	3.19 (1.47, 6.91)	3.46 (1.65, 7.26)	3.42 (1.65, 7.08)
3rd tertile	2.41 (1.17, 4.96)	2.26 (1.09, 4.69)	2.27 (1.09, 4.70)

Model 1: adjusted for age and sex; Model 2: Model 1 + adjusted for household income, education level, occupation, smoking status, drinking habit, body mass index, and physical activity; Model 3: Model 2 + adjusted for consumption of high-lead-containing food.

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
