# Peer review of "Blood Lead Concentrations and Mortality in Korean Adults: the Korea National Health and Nutrition Examination Survey with Mortality Follow-Up"

_ijerph, 2020, doi:10.3390/ijerph17186898_

Round 1
Reviewer 1 Report
The manuscript is concerned with the “Blood lead concentrations and mortality in Korean adults: the Korea National Health and Nutrition Examination Survey with mortality follow-up”. The manuscript, in its present form, has an introduction, methods and entire manuscript in general well written. It is an interesting work within the scope of IJERPH, with high scientific value as a tool for lead exposure and associated mortality. Besides, this manuscript presents a clear description of the work’s conception, methods, and results. The conclusion section should be more comprehensive. Nevertheless, this manuscript has the potential to be a valuable piece of work. Just minor revisions to the following points should be undertaken to justify the recommendation for publication.
Specific comments:
Line 60: consider removing the word “the”.
In table 1, in blood lead level is <2.71 or >2.71?
Line 169: “morality” or mortality
Conclusion: The conclusions should be more scrutinized. It is also important to frame this work within the sustainable development goals of the 2030 agenda, even mentioning the importance of mitigation measures in seafood and cereals consumption for human health.
Author Response
Response to Reviewer 1 Comments
Point 1: Line 60: consider removing the word “the”.
Response 1: Thank you for your kind indication and let us apologize careless mistake. The word “the” was removed.
Point 2: In table 1, in blood lead level is <2.71 or >2.71?
Response 2: We thank the reviewer for catching this typo. It is “>2.71”
Point 3: Line 169: “morality” or mortality
Response 3: The typo was corrected.
Point 4: Conclusion: The conclusions should be more scrutinized. It is also important to frame this work within the sustainable development goals of the 2030 agenda, even mentioning the importance of mitigation measures in seafood and cereals consumption for human health.
Response 4: Thank you for your suggestion. We have added the following sentences to the conclusion (Line 277-282 of the revised manuscript). “While the BLL in most of the study population was under the CDC reference value, continuous efforts will be needed to reduce the BLL in the general population, such as strengthening the standard for lead in food or ambient air. Policies to manage the BLL will contribute to achieving the Sustainable Development Goal target (3.9 By 2030, substantially reduce the number of deaths and illnesses from hazardous chemicals and air, water and soil pollution and contamination) in the long run.”
Reviewer 2 Report
This paper examines the age old contaminant lead and its associated mortality among adults in Korea.
Data was drawn from the KNHANES study.
I thought the study goal was well outlined.
The methods section was also well described.
Overall, the results represent an accurate representation of the methods and statistical analyses performed.
I am glad that some of the limitations of the study was outlined.
Author Response
Response to Reviewer 2 Comments
No comment required to respond
Reviewer 3 Report
This study tried to examine the associations between mortality and blood lead concentrations for adults in Korea using Korea National Health and Nutrition Examination Survey (2008-2013) linked Cause of Death data, which are followed by 2018. The manuscript was written well and presented logistically. The topic is interesting to the readers. My concerns are listed as follow:
- Line 87-88, The analyses of BLLs should be introduced in detail, including the QA/QC information. These information should at least be provided in the form of supplementary materials.
- Line 87-88, Smoking status and Drinking frequency were classified too roughly, can them be described as quantitative or semiquantitative data?
- Line 89-91, Levels of physical activity were classified into two levels, which is also too roughly.
- Line 231-234, These discussion and reference are not of much help to illustrate or support this article.
- Is the Figure under Line 251 the Graphic Abstract? This Figure doesn't fit right here.
Author Response
Response to Reviewer 3 Comments
Point 1: Line 87-88, The analyses of BLLs should be introduced in detail, including the QA/QC information. These information should at least be provided in the form of supplementary materials.
Response 1: We additionally described the QA/QC information in line 86-93 of the revised manuscript.
Point 2: Line 87-88, Smoking status and Drinking frequency were classified too roughly, can them be described as quantitative or semiquantitative data?
Point 3: Levels of physical activity were classified into two levels, which is also too roughly
Response 2&3: The original data used in this study is not in our possession and can only be analysed within the data analysis room of the Korean Center for Disease Control. However, due to COVID 19 situation, the analysis room is not available until this month (Sep. 2020).
I agree that the categorizations of the variables (smoking, drinking, physical activity) are rough, but those categorizations have commonly used in the previous studies. Given that the adjustment of individual socioeconomic status and health behaviors have resulted in a change of less than 10% in the effect estimates (see Table 2), we guess that semiquantitative categorization is unlikely to have a significant impact on the results. In addition, we could lose some samples by using additional information to quantify the variables.
Nevertheless, if you decide that further analysis is necessary, we will ask the editor to extend the revision deadline.
Point 4: Line 231-234, These discussion and reference are not of much help to illustrate or support this article.
Response 4: We tried to discuss the limitation of the blood lead as an indicator of lead exposure in our study. It seems that this part was not clearly described, so we modified the paragraph (Line 246-253 of the revised manuscript). If there is any problem other than just the way it is described, please let us know so that we can improve the manuscript.
Point 5: Is the Figure under Line 251 the Graphic Abstract? This Figure doesn't fit right here.
Response 5: Yes, the figure is the Graphical Abstract. We guess the editorial office has placed the figure in that position.
Reviewer 4 Report
Aside minor typographical errors, the paper was well constructed. As highlighted in the conclsuion the limited number of death cases has imopacted on the conclusion drawn.
The paper latched on already gathered historical data by a third party and in its analysis, clear method was presented (flow chart) around the selection of the subjects included in the study as well as providing detailed statistical analysis applied to inform the study findings.
Results presented were succinct and did highlight on the major outcomes contain in each table/figure.
The weak link in the paper was the introduction section, as very limited study in Asia (6) was used to establish the study rationale.
This can be improved to help strengthen the introduction.
In addition while it is acknowledged effort has been placed at improving the paper language and structure, however there is the need for a proof read to ensure the typo still contain in the work is amended.
Also, it is not clear why the last figure in between line 251 and 252 was placed there without making reference to same in the text as well as provision of figure running title. This also will need amending
Author Response
Response to Reviewer 4 Comments
Point 1: The weak link in the paper was the introduction section, as very limited study in Asia (6) was used to establish the study rationale.
Response 1: For the rationale of this study, in addition to the lack of studies in Asia (especially in Korea), we also suggested the consideration of potential confounding by a dietary factor that has not been examined in the previous studies (Line 57-63 of the revised manuscript).
Point 2: In addition while it is acknowledged effort has been placed at improving the paper language and structure, however there is the need for a proof read to ensure the typo still contain in the work is amended.
Response 2: Thank you for your careful reading of our manuscript. This manuscript has been revised carefully.
Point 3: Also, it is not clear why the last figure in between line 251 and 252 was placed there without making reference to same in the text as well as provision of figure running title. This also will need amending
Response 3: The last figure is the Graphical Abstract. We guess the editorial office has placed the figure in that position.
Round 2
Reviewer 3 Report
The authors have made good modifications according to the comments, and most of my concerns have been resolved. The revised version of the manuscript is a great improvement over the original. I think the manuscript is currently available for publication in IJERPH.